# Morphological Correlates of TRPV1 Agonist-Induced Activation and Defunctionalization of Nociceptor Neurons

**DOI:** 10.3390/ijms262110350

**Published:** 2025-10-24

**Authors:** Gábor Jancsó, Mária Dux, Péter Sántha

**Affiliations:** 1Department of Anatomy, Histology and Embryology, University of Szeged, Kossuth Lajos Sugárút 38, H-6724 Szeged, Hungary; gaborjancso@yahoo.co.uk; 2Department of Physiology, University of Szeged, Dóm tér 10, H-6720 Szeged, Hungary; dux.maria@med.u-szeged.hu

**Keywords:** capsaicin, resiniferatoxin, TRPV1, pain, analgesia, degeneration, regeneration, ganglioside

## Abstract

Transient receptor potential vanilloid type 1 (TRPV1) agonist-induced analgesia is a current hot topic of pain research and a promising possibility to alleviate chronic/neuropathic pain. Local applications in humans and animals and systemic administration in experimental animals of TRPV1 agonists have been demonstrated to produce a long-lasting blockade of nociceptors leaving the function of other types of sensory nerves, as well as autonomic and motor nerve fibers, intact. Morphological studies revealed that TRPV1 agonist-mediated drug action is linked to distinct structural alterations involving reversible and/or irreversible neuronal degenerative processes. This review is intended to summarize the available information on morphological changes associated with TRPV1 agonist-induced activation and defunctionalization of nociceptors expressing the TRPV1/capsaicin receptor. In addition, morphological alterations associated with some pathologies involving TRPV1-expressing nociceptors will also be dealt with. Activation and defunctionalization can be elicited from any domain of TRPV1 receptor-expressing neurons. Considering the similar membrane properties of perikarya, axons and peripheral receptive nerve endings, the term chemosensitive nociceptor neuron is proposed to denote this particular class of primary sensory neurons.

## 1. Introduction

Chemosensitive primary sensory neurons (CPSNs) comprise a unique population of nociceptors with multifaceted functional characteristics and distinct morphological traits. Their involvement in pain mechanisms has put the study of these nociceptors at the forefront of investigations dealing with the physiology and maladaptive responses of the somatosensory system. In this review, we point to the significance of morphological findings which have shaped the development of research in the pain field. These include the morphophysiological characterization of nociceptive primary sensory neurons, which are exquisitely sensitive to capsaicin and bind resiniferatoxin. Studies that aimed to reveal the molecular basis of capsaicin sensitivity have led to the discovery of the transient receptor potential vanilloid type 1 (TRPV1) receptor. Hodological studies utilizing the choleratoxin B subunit drew attention to the role of ganglioside GM1, contained in membrane lipid rafts, in the regulation of TRPV1 activation and expression in CPSNs. Functional morphological studies also revealed the localization, distribution and possible functional significance of the TRPV1 receptor within the different domains of the CPSNs. By synthesizing data from previous observations, we propose and present evidence for a novel concept of the chemosensitive nociceptor.

## 2. Chemosensitive Primary Sensory Neurons: A Unique Class of Nociceptors

CPSNs consist of a particular class of nociceptive primary sensory neurons which express the TRPV1/capsaicin receptor. The term chemosensitive is preferred to capsaicin-sensitive, since (1) these neurons transmit impulses generated by noxious agents of widely different chemical structures also acting through other receptors than TRPV1, and (2) the TRPV1 agonist-induced defunctionalization blocks the activation of CPSNs not only by vanilloid compounds, but also by other chemical irritants [1,2]. Although the irritant property of capsaicin was known for centuries, the fundamentals of the pharmacology of capsaicin were laid by Nicholas Jancsó in the late 1940s. Studying the phlogogenic and pain-producing effects of capsaicin, he discovered that repeated local application of capsaicin or a single high-concentration application to the skin, cornea, mucous membranes and airways resulted in analgesia to pain-producing chemicals, but not mechanical stimuli. This phenomenon has been termed capsaicin desensitization. In addition, neurogenic inflammation, a tissue reaction elicited by chemical irritants acting selectively on nociceptive sensory nerve endings or by the antidromic stimulation of sensory nerves, was also inhibited by capsaicin desensitization.

Early electron microscopic investigations demonstrated long-term mitochondrial swelling in type B sensory ganglion neurons after systemic capsaicin treatment in rats [3,4]. Histochemical studies demonstrated depletion of fluoride-resistant acid phosphatase (FRAP), an enzyme specifically localized in small primary sensory neurons, from terminals of CPSNs projecting to the substantia gelatinosa: an important relay station in the processing of nociceptive information in the spinal cord dorsal horn [5,6]. Based on the similarity of the histochemical localizations of FRAP and substance P in the spinal dorsal horn, the effect of capsaicin treatment on substance P was examined. Immunohistochemical investigations revealed that treatment with capsaicin resulted in a notable depletion of the peptide from the spinal dorsal horn, suggesting its possible role in pain transmission [7]. Interestingly, a similar conclusion has been reached earlier by showing a decreased level of substance P in the spinal cord after capsaicin treatment using a bioassay [8].

The discovery of the selective neurotoxic action of capsaicin and related pungent agents (sensory neurotoxins) enabled the direct morphological identification and characterization of CPSNs [9,10,11] (Figure 1). Capsaicin given to newborn rats, mice and dogs has been shown to induce the selective degeneration of a well-defined population of small type B primary sensory neurons with mostly unmyelinated axons in the trigeminal, vagal and glossopharyngeal sensory ganglia [9,12]. Electron microscopic investigations revealed successive degenerative changes in small sensory ganglion cells, unmyelinated dorsal root and peripheral nerve axons and spinal and brainstem primary afferent terminals. Small type B sensory ganglion cells exhibited marked degenerative changes in swollen mitochondria, with disorganization of their cristae, dilatation of the rough endoplasmic reticulum and marked cytoplasmic osmiophilia [9,13,14]. Considering the time span of the degeneration process, it was considered to be a special, chemically induced form of Cajal’s primary centrifugal degeneration, which is “brought about in the two branches of the sensory cell by granular destruction or by a grave lesion of the soma” [15]. Silver impregnation studies revealed that their spinal and cranial primary afferents project to Rexed’s laminae I and II and, in the case of visceral afferents, to Rexed’s laminae I, V and X of the spinal dorsal horn. In the brainstem, chemosensitive primary afferents project to the subnucleus marginalis and gelatinosus of the trigeminal nucleus caudalis, the paratrigeminal nucleus, the nucleus oralis, the nuclei of the solitary tract and the area postrema [16,17,18,19]. Subsequent studies confirmed and extended these observations by also demonstrating axon terminal degeneration in some other areas of the central nervous system, unrelated to the known projection territories of primary afferent neurons [20]. The possible functional significance of these latter neuronal systems remained largely unclear, but capsaicin-sensitive mechanisms have been explored in the hypothalamic preoptic area [21,22,23], the locus ceruleus [24], the parabrachial nuclei and the hippocampus [25]. In the periphery, chemosensitive afferent nerves have been demonstrated in many organs following systemic or local application of capsaicin or after exposure to capsaicin in vitro. Chemosensitive nerves identified by TRPV1 immunohistochemistry, or by virtue of capsaicin-induced degenerative structural changes or depletion of sensory neuropeptides contained in this particular class of CPSNs were localized, among others, in the skin, the meninges, and the respiratory, cardiovascular, gastrointestinal and urogenital systems [12,26,27,28,29].

Animals treated with capsaicin as neonates develop a life-long analgesia against chemical irritants and a reduced sensitivity to noxious heat. In addition, capsaicin treatment resulted in a complete abolition of the neurogenic inflammatory response, i.e., vasodilatation and plasma extravasation elicited by chemical irritants or antidromic nerve stimulation [9,30,31,32]. Grown up rats treated with capsaicin as newborns have become reliable and widely used models for studies on pain mechanisms and a variety of functions mediated, at least in part, by CPSNs. Prior to the development of genetic knock outs [33,34], neonatal capsaicin treatment offered a possibility to pharmacologically “knock out” the capsaicin receptor, as well as the nociceptive primary sensory neurons. The review of the vast literature on the role of CPSNs in the mechanisms of physiological functions and pathologies involving CPSNs is beyond the scope of the present review, but can be found in comprehensive overviews [12,35,36,37,38,39,40,41,42,43,44,45,46,47,48,49].

The identification and cloning of the capsaicin receptor, termed the transient receptor potential vanilloid type 1 receptor (TRPV1) by Julius and colleagues [50] was a major milestone in the understanding of the molecular mechanism of the action of capsaicin. This discovery opened the way for investigations into the physiological roles of this receptor in nociceptive functions and in a variety of cellular functions and diseases [33,34,50,51,52]. Autoradiographic studies using [^3^H] resiniferatoxin binding [53] and an immunohistochemical demonstration of the vanilloid receptor type 1 (VR1)/TRPV1 receptor [54,55,56] showed similar localization of the capsaicin receptor, mostly confined to the central projection areas of primary sensory neurons. The distribution of the capsaicin receptor coincided with the spinal and brainstem localization of primary afferents that are sensitive to the neurotoxic effect of capsaicin [16].

A comprehensive study on the distribution of VR1 mRNA and VR1-like immunoreactivity revealed widespread localization in the rat and human brain, including primary sensory neurons and some other neurons: for example, in the cerebral cortex and the hippocampus. Importantly, neonatal capsaicin treatment depleted VR1 mRNA from the spinal nucleus of the trigeminal nerve, but not from other areas [57,58]. The localization of sensory neuropeptides including substance P, somatostatin, calcitonin gene-related peptide (CGRP), galanin and vasoactive intestinal polypeptide has also been demonstrated in CPSNs [56,59,60,61,62,63].

Classification of mouse primary sensory neurons using large-scale single-cell RNA sequencing revealed a more complex expression profile of nociceptive neurons, including ion channels, trophic factors, neurotransmitters and neuropeptides, but TRPV1 expression was a major trait of non-peptidergic and peptidergic nociceptive primary sensory neurons [64,65,66,67,68]. In the rat, 85–94 percent of unmyelinated axons in the lumbar dorsal roots disappeared after neonatal capsaicin treatment, suggesting that the overwhelming majority of C-fiber nociceptive sensory ganglion cells are capsaicin-sensitive, i.e., express the TRPV1 receptor in this species [69,70]. Similarly, 64–70 per cent of unmyelinated axons were depleted from the saphenous and sural nerves after neonatal administration of capsaicin [9,71,72]. Since in rat peripheral nerves, the proportion of postganglionic sympathetic unmyelinated axons amounts to about 20 per cent [73], the actual loss of unmyelinated sensory axons may be even greater than 70 per cent after neonatal capsaicin. In contrast, in the mouse, only a reduction by about 41–75 per cent in the number of lumbar dorsal root unmyelinated axons was observed after neonatal capsaicin [74]. In line with these observations, classification of sensory ganglion cell types by applying in situ hybridization [75,76,77] and transcriptomic analysis [78] revealed significant species differences, even between closely related species, such as the mouse and the rat. As compared with the mouse, TRPV1 expression was markedly and significantly higher in the human dorsal root ganglia, as assessed with RNAscope in situ hybridization (human: 74.7%, mouse 32.4%) [75]. Hence, possible species differences should be carefully considered in the interpretation of lesion-induced changes in gene expression and the translatability of these findings into human medicine.

Already, early investigations into the cellular mechanism of the neurotoxic action of capsaicin have indicated a critical role of calcium in this process. Histochemical, electron microscopic, cytochemical and radiochemical findings demonstrated a massive increase in intracellular calcium concentration and sequestration of calcium in mitochondria [79,80]. Capsaicin-induced swelling of mitochondria and disorganization of their cristae after systemic administration also indicated the involvement of this cell organelle in the capsaicin-induced impairment of sensory ganglion neurons [3,9]. Subsequent in vitro light and electron microscopic investigations into vagal nodose ganglion neurons confirmed the role of calcium in capsaicin-induced neuronal death and revealed that it can be inhibited by the removal of extracellular calcium [81,82]. Further studies confirmed these observations on cultured dorsal root ganglion neurons by showing the capsaicin-induced influx of ^45^Ca^2+^, which was inhibited by Ruthenium red, indicating the sequestration of calcium into the mitochondria [82,83]. Capsaicin-induced death of cultured DRG neurons was prevented by the removal of extracellular calcium. Capsaicin also resulted in the uptake of Co^2+^, which can be visualized by a histochemical reaction developed by Hogan [84], and can be inhibited by Ruthenium red [85,86]. Mitochondrial depolarization [87] and changes in mitochondrial permeability [88] have been demonstrated to significantly contribute to the mechanism of capsaicin-induced cell death. The role of calcium-activated proteases, such as calpain in the capsaicin-induced sensory ganglion cell death, was also revealed [85,89]. In addition, capsaicin-induced neuronal death may also be accounted for, at least in part, by an apoptotic process involving the caspase cascade [90,91].

Capsaicin-induced cell death has been shown to be mediated by the activation of the TRPV1 receptor in TRPV1-transfected HeLa cells. At relatively low concentrations (1 µM), capsaicin induced a fast and transient increase in intracellular Ca^2+^, leading to membrane depolarization, impairment of plasma membrane integrity and finally, cell death without mitochondrial dysfunction. This latter observation suggests differences in mitochondrial function between TRPV1-transfected HeLa cells and sensory ganglion neurons, since in nociceptive sensory neurons, mitochondrial mechanisms play a fundamental role in capsaicin-induced cytotoxicity [92]. At a high concentration (100 µM), capsaicin resulted in a TRPV1-independent cell death, inducing a persistent increase in intracellular Ca^2+^, mitochondrial dysfunction, plasma membrane depolarization and disruption of membrane integrity [93].

Neuronal degeneration induced by capsaicin provided a significant model for the study of cellular processes involved in cell death. Previous studies demonstrated that an intracellular increase in Ca^2+^ is involved in both the capsaicin- and glutamate-induced neuronal cell death [80]. Recently, a detailed analysis of the mechanisms of excitotoxicity, including capsaicin-induced neurotoxicity, confirmed the fundamental role of calcium and reactive oxygen species in cell death and, importantly, revealed the crucial role of the mitochondrial electron transport chain in this process [92]. It has been demonstrated that the expression level of the mitochondrial electron transport chain components critically regulates the cellular response to excitotoxic agents, such as capsaicin. Interestingly, it has been shown that low levels of the components of the electron transport chain protect nociceptor neurons from capsaicin-induced toxicity by mitigating cellular calcium overload and generation of mitochondrial reactive oxygen species. The findings may also promote the understanding of the mechanisms of the neuropathological changes associated with diabetes, inflammation and chemotherapeutic agents.

## 3. Morphological Correlates of Activation of CPSNs

Histochemical studies on the effect of capsaicin on sensory ganglion neurons in newborn rats provided the first evidence for the involvement of calcium ions in the molecular mechanism of the action of capsaicin [79]. Subsequent studies further supported the role of calcium in capsaicin’s actions. Following prior administration of ^45^Ca^2+^, capsaicin given to newborn rats resulted in a rapid increase in radioactivity in the sensory ganglia. Autoradiography revealed a marked accumulation of ^45^Ca^2+^ in small sensory ganglion neurons (see Figure 1). Electron microscopic histochemistry and X-ray microanalysis disclosed the mitochondrial accumulation of Ca^2+^ in small, type B ganglion cells [80].

A histochemical method based on the visualization of capsaicin-induced cobalt uptake [84], and calcium imaging have become widely used techniques to demonstrate the activation of CPSNs and the TRPV1 receptor. For a visualization of activated CPSNs, calcium imaging is an obvious choice. This approach was applied to identify sensory ganglion cell clones, which express the capsaicin/TRPV1 receptor [50]. For morphological studies, however, a combination of the cobalt uptake assay with, for example, immunohistochemistry, has been shown to be a reliable approach to examine separate populations of sensory ganglion neurons [83,94]. Different subpopulations of dorsal root ganglion neurons activated by diverse cutaneous noxious chemical stimuli have been identified by using activating transcription factor 3 immunohistochemistry [95].

Capsaicin-induced activation of peripheral axons and nerve endings of CPSNs has also been visualized with calcium imaging. Calcium signals were detected in the corneal nerves of mice by using a fluorescent Ca^2+^ indicator or injection of viruses carrying a genetically encoded calcium indicator and red fluorescent protein [96,97]. Ultrafast optical recording of capsaicin-induced calcium and sodium ion movements revealed topographical differences along single nociceptive axons of cultured sensory ganglion neurons. In the most terminal portions of the neurites, signal generation is not mediated by voltage-gated sodium channel activation but by Ca^2+^ signaling involving TRPV1 channels [98]. The TRPV1 channel-mediated increase in axonal and terminal Ca^2+^ also results in the release of neuropeptides, which are involved in various local tissue reactions [12,39,99]. Immunohistochemical studies demonstrated structural alterations and the release of CGRP from corneal nerve fibers, following stimulation with capsaicin. Capsaicin exposure resulted in an increase in CGRP-immunoreactive axonal varicosities and a decrease in nerve fiber density and β-tubulin immunoreactivity [100]. These findings have significant clinical relevance in assessing the progression of neuropathic changes resulting from disorders such as diabetes and treatment with chemotherapeutic agents [101,102].

Immunohistochemical localization of phosphorylated ERK1/2 has also been applied to reveal the activation of CPSNs, following the perineural application of capsaicin [103]. In addition, an intracutaneous injection of capsaicin induced the phosphorylation of ERK1/2 in cutaneous nerves and nerve endings [104]. Similarly, ERK-immunohistochemistry revealed the activation of spinal dorsal horn neurons following stimulation by capsaicin in the central terminals of CPSNs [105]. In recent years, the molecular morphology of the TRPV1 channel has also been revealed by applying electron cryo-microscopy [106,107]. By applying this technique, the molecular determinants of the activation of this ion channel have also been elucidated [108]. A detailed account of these observations is beyond the scope of the present review.

## 4. Structural Changes in C-Fiber Nociceptive Afferents Underlie the Long-Lasting Analgesia Induced by Capsaicin

Perineural application of capsaicin has been shown to produce a selective, long-lasting, at least of a one-year duration, chemical and thermal analgesia, strictly confined to the innervation territory of the treated peripheral nerve [109,110,111,112]. In addition to the analgesic effect, perineural capsaicin completely abolished neurogenic plasma extravasation in the affected skin area [111,112] and markedly inhibited neurogenic sensory vasodilatation [113] and reactive cutaneous hyperemia [114]. Immunohistochemical studies revealed a profound reduction in epidermal axons, as assessed by staining with antibodies against protein gene product 9.5, a pan-neuronal marker, and the neuropeptides CGRP, substance P and somatostatin [115,116]. Cutaneous sensory nerve endings that were immunopositive for RT97, a marker of capsaicin-insensitive sensory nerves, remained unaffected by capsaicin treatment [116]. Further, neurochemical and histochemical studies revealed an inhibition of the axoplasmic transport of endogenous proteins, for example, FRAP (identical to thiamine monophosphatase [TMP]), exogenous proteins, like horse radish peroxidase [117], and the neuropeptides substance P and somatostatin [118]. Quantitative immunohistochemical studies revealed marked reductions of 61, 46 and 23 percent in the proportions of small-sized dorsal root ganglion neurons expressing substance P, somatostatin and CGRP, respectively, at least up to three months after perineural capsaicin treatment. Importantly, the proportion of LA4-immunoreactive neurons, which express lactoseries carbohydrate epitopes [119] and essentially do not co-localize with neuropeptides, was also markedly reduced, indicating that perineural capsaicin treatment affected non-peptidergic nociceptive CPSNs as well [120].

Although the exact mechanism of the action of perineurally applied capsaicin is still unclear, downregulation of nociceptive ion channels resulting from an inhibition of the retrograde axoplasmic transport of NGF, necessary for the expression of TRPV1 and other nociceptive ion channels [55,121,122,123], may significantly contribute to capsaicin’s analgesic action. A significant decrease in the number of TRPV1-immunoreactive neurons and a transient decrease in the neuronal expression of TRPV1 mRNA have been demonstrated in the sensory ganglia, relating to a peripheral nerve treated with capsaicin [124]. The quantitative morphometric data, showing that only about 30 per cent of unmyelinated axons are lost after perineural capsaicin [120], suggest that capsaicin may differentially affect subpopulations of chemosensitive afferents, which amount to about 70 per cent of unmyelinated axons in peripheral nerves [9,71,72]. Further investigations are needed to disclose whether differing molecular traits of subpopulations of TRPV1-expressing nociceptive neurons [64] may explain the disparate responses to perineural capsaicin. Differential expression of electron transport chain components by subpopulations of primary sensory neurons may also contribute to the different resilience of TRPV1-expressing neurons to capsaicin [92]. Further, a capsaicin-induced increase in calcium influx in sensitive neurons may activate pathways that are involved in neuronal/axonal degeneration [79,81,92].

Considering the long-lasting selective analgesic and antihyperalgesic actions of perineurally applied capsaicin, the possible therapeutic application of this approach has long been proposed [110,125]. More recently, the use of resiniferatoxin, an ultrapotent agonist of the TRPV1 receptor, has been suggested as an alternative to capsaicin. Perineural administration of RTX has been reported to produce thermal and chemical analgesia with a very low level of toxicity. At the very low concentrations of 0.0001% and 0.001%, resiniferatoxin induced marked thermal analgesia lasting for at least 1 and 14 days, respectively. The response to noxious pressure was also reduced, albeit only for a shorter period of 1–3 days [126,127]. Importantly, the number of degenerating unmyelinated axons amounted to less than one percent in the treated nerves [126]. By contrast, many unmyelinated axons displayed marked changes 1–2 days after perineural capsaicin (0.25–1.0%), consisting of swelling and the accumulation of axoplasmic organelles, which are indicative of the inhibition of axoplasmic transport [111,118,128]. Two weeks after treatment, unmyelinated axons appeared to be closely packed together without intervening Schwann cell processes, which normally separate them [117]. Furthermore, two independent research groups have found about a 30% loss of unmyelinated axons in the treated peripheral nerves 6–8 weeks after capsaicin treatment [120,129].

The time span of the degeneration process after perineural capsaicin may provide a possible explanation of these apparently contrasting findings. The marked loss of unmyelinated axons after perineural capsaicin has been attributed to a dying back type-delayed degenerative process [120]. Hence, examination of the resiniferatoxin-treated nerves only a few days later may not reveal changes in axon numbers. Capsaicin and resiniferatoxin exert similar neurodegenerative changes in the newborn [9,14,16,74] and adult [17,130,131] animals. It is worthy of note that the development of capsaicin-induced long-lasting analgesia was associated with sensory (nociceptive) nerve fiber loss after the perineural [110,111,112,117,120,129], epicutaneous [132,133] and intradermal application of capsaicin [133]. The effects of vanilloids following different routes of administration are summarized in Table 1.

These findings have led to the development of a capsaicin patch (Qutenza, Averitas Pharma Inc.) containing capsaicin at the very high concentration of eight percent, which is now used to alleviate postherpetic and also diabetic neuropathy-induced pains in humans.

Perineural treatment with capsaicin also elicited profound structural and neurochemical changes in the dorsal horn of the spinal cord. Perineural capsaicin treatment resulted in depletions of the neuropeptides subsatnce P and CGRP are neuropeptides, TRPV1 is not; isolectin B4 is a plant lectin which binds to a cellular glycoprotein in a highly selective way [118,134,135,136,137,138].

In adult rats, systemic administration of capsaicin produced marked degeneration argyrophilia in the marginal zone and the substantia gelatinosa, i.e., in Rexed’s laminae I and II, indicating the degeneration of chemosensitive primary afferents [130] which express the TRPV1 receptor [139]. Prior perineural capsaicin treatment abolished degeneration argyrophilia within the spinal projection territories of the capsaicin-treated nerve, producing a gap in the continuous silver-stained band in lamina I and II [120,140]. It is still not entirely clear whether this "capsaicin gap" may be accounted for by a loss (degeneration) of C-fibers or a change in the capsaicin sensitivity of C-fiber primary sensory neurons: for example, due to the lack of retrogradely transported NGF, which is fundamental for the expression of the vanilloid receptor [55,122,141,142,143,144]. The observations that the number of unmyelinated nerve fibers decreased by 30–35 per cent in the capsaicin-treated (saphenous) nerve [120,129] and that small dorsal root ganglion cells are selectively and substantially lost after perineural capsaicin [120], strongly suggested that the capsaicin gap may be largely accounted for by a loss of CPSNs.

Perineural treatment with capsaicin has also been shown to inhibit collateral sprouting from intact nociceptive axons, innervating the adjacent skin area into the chemodenervated skin [113,145,146]. Furthermore, transection of the capsaicin-treated nerve resulted in marked reinnervation of the chemodenervated skin by collateral sprouting of axons, which innervated the adjacent skin area. This implies that perineural capsaicin treatment of intact nerves innervating the skin area adjacent to the denervated skin may inhibit collateral sprouting from that nerve, and thus may impede the development of neuropathic pain resulting from the sprouting of intact sensory nerves into the denervated skin [147,148,149].

Besides inducing degenerative changes in CPSNs, capsaicin has also been shown to promote neurite outgrowth of sensory neurons. Prior systemic administration of capsaicin to adult rats resulted in an increased neurite outgrowth of cultured dorsal root ganglion neurons [150]. Recent findings demonstrated that activation of the TRPV1 receptor may enhance the propensity of nociceptive sensory neurons to regenerate. Activation of the TRPV1 receptor by capsaicin promoted axonal regeneration of cultured sensory ganglion cells. This effect was dependent on the TRPV1 receptor function and was restricted to nociceptive dorsal root ganglion neurons. In in vivo experiments, prior perineural capsaicin treatment of the injured sciatic nerve also promoted axonal regeneration. An increased Ca^2+^ influx and the consequent activation of signaling pathways involving protein kinase A and CREB have been shown to be involved in the mechanism of the pro-regenerative effect of capsaicin [151]. Further studies are warranted to resolve the apparent contrasting findings of these and earlier studies showing the permanent loss of dorsal root ganglion neurons after systemic [9,130] or perineural capsaicin treatment [120,129] and the failure of nerve regeneration after perineural capsaicin application [113,146].

Injuries to peripheral nerves, such as axonotmesis and neurotmesis, evoke massive activation and proliferation of microglial cells in the somatotopically related areas of the superficial spinal dorsal horn [136,152,153,154,155]. Interestingly, perineural application of the TRPV1 agonist, capsaicin, which may be regarded as a special (chemical) form of peripheral nerve injury, failed to induce significant spinal microgliosis. Notably, the defunctionalization of CPSNs by prior perineural capsaicin treatment did not influence the development of spinal microgliosis elicited by peripheral nerve transection [136]. These findings are in accord with the observation that the resiniferatoxin-induced blockade of the C-fiber function failed to affect the injury-induced microglial response [156], and collectively suggest that damage to CPSNs is neither sufficient nor necessary for the initiation of spinal microgliosis [136]. Available experimental evidence indicates that the nerve injury-induced spinal microglial reaction is evoked by the injury of large Aß primary afferents [136,156,157].

**Table 1 ijms-26-10350-t001:** The in vivo effects of capsaicin and other vanilloids upon different routes of applications.

Type of Treatment	Acute Changes	Chronic Changes
Neonatal systemic 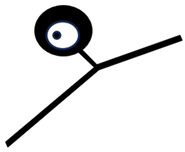	Depolarization of dorsal root and peripheral nerve axons [158,159] Ca^2+^ accumulation in small sensory ganglion cells [79,80] Degenerative changes in small DRG neurons [9,13,14] (including mitochondrial swelling) Degeneration of unmyelinated dorsal root and peripheral nerve axons [9,13,14] Degeneration of spinal and medullary primary afferent terminals [10,16]	Long-lasting decreased sensitivity to noxious mechanical, chemical and heat stimuli [9,12,38,160,161,162,163,164] Loss of neurogenic inflammation [9,30,165] Reduced thermal hyperalgesia [166,167] Decreased visceral sensitivity [168] Loss of B-type sensory ganglion cells (~50% of all DRG neurons) [9,169,170] Loss of C-fiber afferent axons and nerve endings (reduction by 70% and 90% of unmyelinated axons in sensory nerves and dorsal roots, respectively) [9,69,71,72,169,171] Depletion of sensory neuropeptides and specific proteins (e.g., IB4, FRAP/TMP) [56,59,60,61,172,173] Sprouting of spinal myelinated afferents [174,175,176]
Adult systemic 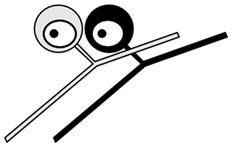	Ca^2+^ accumulation in small sensory ganglion cells [81,82,92] Degenerative changes in small DRG neurons (including mitochondrial swelling) [3,81,130,177] Degeneration of unmyelinated dorsal root and peripheral nerve axons [130,178,179,180] Degeneration of spinal and medullary primary afferent terminals [17,20,130]	Decreased sensitivity to chemical irritants and heat [1,2,181] Reduced neurogenic inflammation [1,2,9,32,181,182,183] Decreased visceral sensitivity [99] Depletion of sensory neuropeptides and specific proteins (e.g., IB4, FRAP/TMP) [5,7,55,61,170] Degeneration of small sensory ganglion cells (~17% of all DRG neurons) [81,130,177,184] Loss of C-fiber sensory axons and nerve endings (reduction by 30–50% of unmyelinated axons in sensory nerves) [130,184]
Local application 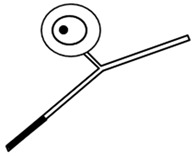	Burning pain, [2,185,186] Vasodilatation—Axon reflex flare [2,185,186] Hyperalgesia [2,185] Block of action potential initiation/conduction	Increased noxious heat threshold [186] Chemoanalgesia [2] Reduced neurogenic inflammation [187] Degenerative changes in peripheral C-fiber sensory axons and nerve endings (in part reversible) [132,133,187] Axoplasmatic transport block (?) Loss of thermal hyperalgesia Decreased visceral sensitivity [188] Depletion of sensory neuropeptides from sensory nerve terminals [132,133,189] Regeneration of cutaneous sensory nerves [132,133] Therapeutic effect in certain types of europathic pain (Qutenza) [190,191]
Intrathecal or intra- cisternal application 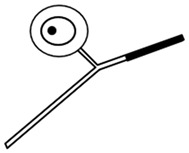	Pain [192,193,194] Chemoanalgesia [193] Cutaneous vasodilatation [193,194] Mechanical allodynia [193].	Chemoanalgesia [192,193] Increase in noxious heat threshold [192] Inhibition of heat hyperalgesia [195,196] Degeneration of spinal/medullary primary afferent terminals [193,194] Depletion of sensory neuropeptides and specific proteins (e.g., Substance P, IB4, FRAP/TMP) from central but not from peripheral branches of DRG neurons [192,194] Preserved cutaneous neurogenic inflammation [194,197] Possible therapeutic application [191,195,197,198,199]
Perineural or local nerve application 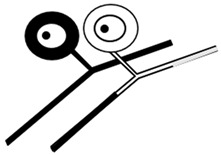	Depolarization of C-fiber afferents [47,81,158,200,201,202] Action potential conduction block [201,202,203,204,205,206] Block of axoplasmic transport [110,111,118,143]	Selective regional chemical and thermal analgesia [110,112,206] Loss of neurogenic plasma extravasation [110,115,146,207] Reduced neurogenic sensory vasodilatation [113,114] Reduced thermal hyperalgesia [208] Reduction by >30% in C-fiber sensory axons and epidermal nerve endings [115,120,129,207,209] Loss by >30% in small DRG neurons [120] Depletion of sensory neuropeptides and specific proteins (e.g., IB4, FRAP/TMP) [118,134,135,136,137,207] Increased expression of injury peptides (galanin, VIP) and GM1 ganglioside in DRG neurons and spinal dorsal horn (phenotypic switch) [135,137,210] Transganglionic degeneration of C-fiber primary afferents [120] Inhibition of C-fiber collateral sprouting of intact cutaneous afferents [113,146]

## 5. Localization of GM1 Ganglioside in CPSNs and Its Functional Significance

The choleratoxin B subunit (CTB), or its horseradish peroxidase conjugate (CTB-HRP), has been shown to be transported by myelinated primary afferents into the deeper layers of the spinal dorsal horn following an intraneural injection [211]. Injection of CTB into an injured peripheral nerve resulted, in addition to the deeper layers of the dorsal horn, in an intense labeling of the superficial dorsal horn, the substantia gelatinosa, too [212]. This phenomenon was attributed to central sprouting of injured myelinated primary afferents and it was proposed to be an important mechanism contributing to disturbed sensory processing and the development of neuropathic pain [210,212,213,214,215]. Quantitative histochemical and morphometric studies, however, disclosed that injection of CTB or CTB-HRP into an injured nerve labeled not only A-fiber primary sensory neurons, but also small C-fiber sensory ganglion neurons [216,217]. The direct electron microscopic histochemical demonstration of CTB-HRP in unmyelinated dorsal root axons indicated that nerve injury-induced CTB-HRP labeling of the substantia gelatinosa may be attributed to a phenotypic switch of C-fiber nociceptive CPSNs, rather than the sprouting of Aß primary afferents [216,217,218,219,220,221,222,223]. Marked sprouting of myelinated primary afferent fibers has been demonstrated in the spinal dorsal horn of rats treated with capsaicin as neonates [174,175,176].

Perineural capsaicin treatment produces a selective regional analgesia to heat and chemical irritants in the innervation territory of the affected nerve [110,111,206,224]. Interestingly, perineural capsaicin treatment of the sciatic nerve also resulted in a robust transganglionic labeling of the substantia gelatinosa with CTB-HRP [210], due to the uptake and transport of CTB-HRP by the affected C-fiber CPSNs [135]. Since CTB selectively binds the GM1 ganglioside [211,225,226], these observations prompted investigations into the possible role of this ganglioside in the function of CPSNs.

Pharmacological manipulation of membrane ganglioside levels in cultured DRG neurons revealed an important role of GM1 in the regulation of neuronal activation by capsaicin. Inhibition of glucosylceramide synthase (GCS), the rate limiting enzyme of ganglioside synthesis, markedly and significantly reduced the proportion of cultured DRG neurons activated by capsaicin. It has also been demonstrated that gangliosides may also modulate the expression of the TRPV1 receptor protein [227,228,229]. The proportion of CGRP-immunoreactive CPSNs was not reduced significantly, indicating that ganglioside depletion may differentially influence the expression of TRPV1 and CGRP, respectively [228]. Further, ganglioside depletion by inhibition of glucosylceramide synthase markedly reduced the neurite outgrowth of cultured primary sensory neurons. This may, at least in part, result from the reduced functionality of the nerve growth factor (NGF) signaling, since both the GM1 level and lipid raft integrity are essential for the normal signalization by the NGF-TrkA system [230,231]. This notion was also supported by the finding that the acute TRPV1 sensitizing effect of NGF via TrkA activation was reduced by the inhibition of GCS [228].

The complex effect of GM1 on neuronal gene expression was demonstrated in neurons from the brains of genetically modified mice, serving as an animal model of GM1 gangliosidosis. Numerous transcripts regulated by the altered GM1 metabolism were functionally associated with neurodegenerative processes, which are characteristic for Parkinson-, Alzheimer- and Huntington-like neurodegenerative mechanisms [232]. Nuclear GM1 has also been shown to regulate, with an epigenetic mechanism, the expression of ganglioside synthase genes through binding to acetylated histones on the promoters of these genes [233].

The fundamental change in ganglioside metabolism demonstrated following different types of peripheral nerve injuries in rodents and primates [216,217,221] may represent a common cellular response of primary sensory neurons to lesions interfering with their axonal integrity. Although the mechanism of this pathological reaction is unknown, preliminary analysis on the expression of enzymes involved in the cellular metabolism of gangliosides suggest an injury-induced downregulation of enzyme(s) involved in GM1 degradation. The results obtained by using high performance liquid chromatography–mass spectrometric analysis of glycosphingolipid (GSL) content of intact and transected peripheral nerves indicate the accumulation of several ganglioside species in the peripheral axons and possibly in dorsal root ganglia, after a peripheral nerve injury [234]. It is suggested that the accumulation of ganglioside GM1, and possibly other GSLs in injured neurons, may represent a new, hitherto unrecognized form of secondary ganglioside (storage) disorder, affecting injured primary sensory neurons [135,221,228].

## 6. A Novel Concept of the Capsaicin-Sensitive Nociceptive Primary Sensory Neuron: The Chemosensitive Nociceptor

In classical studies, the functions attributed to histologically identified specialized nerve endings were based on insightful consideration, rather than experimental evidence. Observations on cutaneous sensibility led Max von Frey to conclude that the epidermal free nerve endings were responsible for the sensation of pain in the human skin [235]. Early electrophysiological investigations revealed that primary afferents with highly specific sensitivities underlie the complexity of cutaneous sensibility, including pain [236]. Extensive investigations over several decades in the twentieth century provided unambiguous evidence for populations of the primary sensory neurons of highly specific functional characteristics [67,237,238,239]. These observations paved the way for the notion that specific modalities are conveyed by populations of highly specific primary sensory neurons. Hence, the specificity theory, which took its origin in Johannes Müller’s idea of “specific nerve energies”, has become one of the most important concepts of somatosensory sensation. Investigations into the mechanisms of the selective sensitivity to specific stimuli of particular sensory nerves have suggested that specific transducer molecules localized in their terminal membrane may confer specific sensitivities, including that of pain, to the primary afferent endings. The discovery of the highly selective neuroexcitant/neurotoxic action of capsaicin, the pungent principle in hot peppers, on CPSNs [1,2,9,10] paved the way for the molecular identification of the first nociceptive transducer protein, the VR1/TRPV1 nociceptive ion channel [50,240]. Recent electrophysiological studies and investigations applying genetically modified animals and large-scale single cell RNA sequencing provided evidence for specific subtypes of (nociceptive) primary sensory neurons, which express specific combinations of TRPV1, neurotransmitter, neuropeptide and ion channel transcripts [64,66,241].

Pharmacological and electrophysiological studies suggested that, in line with classical concepts of receptor physiology, action potentials in nociceptive afferents are generated at the terminal segments of the sensory nerve, termed free nerve endings or fine sensory endings [242]. In particular, the archetypal nociceptive ion channel, the capsaicin/TRPV1 receptor, which integrates multiple pain-producing stimuli, was localized to the terminal portions of fine sensory endings by light and electron microscopic histochemistry [132,133,243,244,245].

Electrophysiological analysis of the effect of capsaicin on different domains of the sensory neuron indicated that the peripheral and central terminals are apparently more sensitive to capsaicin than axons of the peripheral nerve trunk. It has been suggested that capsaicin’s actions are predominantly mediated by the activation of the terminal portions of the CPSNs [246]. Henceforth, the contribution of other compartments of the sensory neuron, such as the peripheral axons or preterminal axon segments, have not been considered as having similar receptive characteristics and potential for initiating the activation of CPSNs. However, a considerable number of experimental and clinical observations accumulated, which suggest that domains of the CPSN, other than the peripheral nerve terminals, may also be capable of the activation of CPSNs.

The first indication that domains of CPSNs other than the nerve terminal membrane may also be sensitive to capsaicin in in vivo experiments came from studies which demonstrated the long-lasting functional blockade of CPSNs following the application of capsaicin directly onto the peripheral nerve trunk. Perineural capsaicin treatment induced a long-lasting (up to a year) increase in heat–pain sensitivity and a complete inhibition of cutaneous neurogenic inflammation, confined to the innervation territory served by the treated nerve [110,111,112]. A brief application of capsaicin (32–50 mM) onto the cranial and peripheral nerves has also been shown to produce a selective block of the conduction of nerve impulses in C and Aδ axons [201,202,205]. The impairment of impulse conduction in peripheral nerves, as assessed by recording compound action potentials following supramaximal stimulation, is not permanent after perineural capsaicin [202,205]. However, there is a prolonged decline in the ability of C-fiber afferents to excite spinal cord dorsal horn neurons [205]. The long-lasting functional impairments may be explained, at least in part, by the profound and lasting depletion, from the CPSNs, of the neuropeptides substance P and CGRP [118,134], which mediate the neurogenic inflammatory response and are also involved in the mediation of nociceptive signals [247,248,249]. In addition to these mechanisms, the delayed loss of C-fiber afferents and small sensory ganglion neurons may also significantly contribute to the long-term effects of perineurally applied capsaicin [120,129]. Although defunctionalization of the chemosensitive primary afferents appears to be complete and permanent after the perineural capsaicin treatment, it resulted in the loss of only about 30 per cent of all unmyelinated axons, whereas the proportion of unmyelinated sensory axons amounted to about 70 per cent of all unmyelinated afferent nerve fibers in the saphenous nerve [120,129,250]. This observation may point to the possibility that subpopulations of CPSNs may differ in their sensitivity towards the neurotoxic action of capsaicin, probably due to their differing molecular traits. Indeed, separate subpopulations of CPSNs have been revealed by using neuronal subgroup elimination transcriptomics [251] and large-scale single-cell RNA sequencing [64,66].

Further studies revealed a direct excitatory action of capsaicin on nociceptive peripheral sensory nerve fibers. In an in vitro study, it has been demonstrated that application of capsaicin onto the sciatic nerve resulted in the depolarization of C-fiber afferents [200]. This depolarization remained unaffected in the presence of tetrodotoxin. Capsaicin-induced depolarization was eliminated or strongly inhibited in dorsal root preparations obtained from rats that were pretreated with capsaicin to selectively destroy C-fiber primary afferent neurons [200]. Vagal afferent axons [81,252] and dorsal root axons [158,159] have also been shown to be depolarized by capsaicin. Similar observations have been reported in in vivo experiments, following the application of capsaicin to the saphenous and vagal nerves of cats. Capsaicin applied directly onto the nerve trunk induced increased discharge activity in C- and Aδ-, but not A-fiber afferents [201,202]. These observations are in line with the findings, showing depolarization of small C-type neurons with slow axonal conduction, but not A-type DRG neurons with greater axonal conduction velocity [253]. Behavioral responses to intracisternally injected capsaicin also suggested the excitation of dorsal root axons, by provoking protective scratching movements. This was followed by an almost immediate development of corneal chemical analgesia, associated with the rapid degeneration of trigeminal afferents projecting to the subnucleus caudalis. Neurogenic plasma extravasation, however, could be readily induced in the analgetic facial skin, suggesting that the sensory ganglion neuron with its peripheral branch forms an independent functional entity capable of responding to noxious stimuli by secreting vasoactive peptide(s). These findings clearly showed that CPSNs possess a dual function, consisting of the transmission of noxious impulses to the central nervous system (afferent function), and the peripheral release of vasoactive agents (efferent function), which may manifest independently of each other [193,194,197]. The above observations strongly suggested the presence of functional TRPV1 receptors on C-fiber primary afferent axons running in dorsal root and peripheral nerves. Extending previous light microscopic findings [132,133,244,245,254], the TRPV1 receptor has been localized in peripheral unmyelinated axons at the ultrastructural level [255]. TRPV1 undergoes anterograde axonal transport [256,257] and is translocated to the neuronal membrane by an exocytotic process involving ⍺CGRP [258]. Furthermore, it has also been revealed that these axonal receptors are functional, since the application of capsaicin resulted in the release of CGRP from the peripheral nerves [255,259,260]. Importantly, morphological evidence has also been presented for the secretion of CGRP from capsaicin-sensitive unmyelinated afferent nerve fibers [255]. Further independent evidence for the presence of functional TRPV1 receptors on peripheral chemosensitive afferents has been provided by the demonstration of the increased phosphorylation of extracellular ERK1/2 in small DRG neurons, induced by the application of capsaicin onto their relating peripheral nerves [103]. Perineural application of capsaicin has also been shown to activate spinal dorsal horn neurons, as assessed by c-Fos immunohistochemistry. It has been concluded that the capsaicin receptor on the (sciatic) nerve is involved in the transmission of noxious information [261]. Application of capsaicin onto the mouse sciatic nerve has also been shown to induce enhanced axonal outgrowth through the activation of a precondition-like response in a TRPV1 receptor-dependent manner [151]. However, the perineural application of capsaicin prevented collateral sprouting [146] of neighboring intact axons into the denervated skin area. This inhibitory effect was abolished by the transection of the capsaicin-treated nerve [113].

Studies making use of the selective neurotoxic action of capsaicin also indicated that besides peripheral receptor endings, other domains of CPSNs are also sensitive to capsaicin. Intrathecal or intracisternal injection of capsaicin induced lasting analgesia [192,193] and also caused depletion of substance P from the spinal dorsal horn [192]. Intrathecal and intracisternal injection of capsaicin at micromolar concentrations also resulted in the selective degeneration of unmyelinated primary afferent terminals in the spinal dorsal horn [193,194,262]. Similarly, electron microscopic examination of different organs exposed to capsaicin in vitro demonstrated a rapid osmiophilic degeneration of unmyelinated sensory axons [178,179,180].

Histochemical studies on the sensory ganglia obtained from capsaicin-treated newborn rats were the first to demonstrate that capsaicin induces a massive influx of Ca^2+^ into small DRG neurons [79,80]. Calcium influx has been shown to play a fundamental role in the activation [50,83] and also the death of nociceptive primary sensory neurons [79,80,81,85,92].

These observations strongly suggested that TRPV1 receptors are distributed not only at the receptive peripheral (and central) endings, but also in the membrane of the perikarya and the peripheral and central (dorsal root) axons of the nociceptive CPSNs. Importantly, these findings are strongly supported by the observations which demonstrated the uniformity of the vanilloid receptors present at different parts of the primary afferent neuron [263]. Electrophysiological investigations into the actions of capsaicin applied directly onto peripheral nerves also supported this notion; studies utilizing single-fiber recording indicated that “membrane properties of capsaicin-sensitive mechano-heat sensitive units may be similar at the receptive nerve endings and the parent axon” [204]. Direct evidence for the generation of action potentials in peripheral axons by heat stimulation has also been presented. The thresholds and discharge rates of action potential to heat stimulation of the nerve trunk paralleled those recorded after stimulation of the receptive field of the same polymodal nociceptive unit. Moreover, it has been concluded that heat sensitivity and generation of action potentials may be regarded as a normal capacity in peripheral nerves, and involves TRPV1 and possibly other heat-transducing channels, too [264]. The observation that the sensitization of the TRPV1-mediated heat response by inflammatory mediators showed similar characteristics in cutaneous nociceptor endings and peripheral nerve axons, and the lack of sensitization in TRPV1 knock out animals, also support the role of TRPV1 as a transducer of noxious stimuli, which affect the peripheral nerve [259,260]. Studies on the human peripheral nerve disclosed that membrane properties of peripheral nociceptive axons closely resemble those observed in animal experiments by showing heat-induced generation of action potentials and the development of projected pain [260,265].

Recently, it has been revealed that the capsaicin-evoked action potential may be brought about by the depolarization elicited by the TRPV1 activation-induced cation influx, and the anion efflux-mediated depolarization was evoked by the activation of anoctamin 1 (ANO1), a calcium-activated chloride channel [266]. The great majority (~80 per cent) of small sensory ganglion neurons that are immunoreactive for ANO1 are TRPV1-immunoreactive, too [267]. A functional interaction of TRPV1 and ANO1 ion channels has been documented not only at the perikaryal membrane of CPSNs, but also at the central synaptic terminals [266]. In addition, ANO1 is specifically involved in TRPV1 receptor-mediated nociceptive behavior [266]. ANO1 channels are also activated by noxious heat stimuli [267]. The role of ANO1 in nociceptive functions is further supported by findings showing that prostaglandin E2, a key molecule in nociceptor sensitization, elicits sustained spiking activity in axons of DRG neurons and triggers action potentials through the activation of the ANO1 and Nav1.8 ion channels [268].

Axonal TRPV1 receptor activation, by changing the local tissue microenvironment through the release of e. g., CGRP, may provide favorable conditions for the perineural spread and growth of tumors [269,270]. Moreover, CGRP, the most abundant peptide in CPSNs, promoted tumor progression and, in turn, CGRP receptor antagonists inhibited tumor growth. The in vitro observations demonstrating a close anatomical connection between TRPV1-expressing nociceptive nerves and gastric cancer cells support this notion [271].

In conclusion, there are ample morphological, functional and pharmacological data which unambiguously disclosed the presence and functionality of the archetypal nociceptive ion channel, TRPV1, in the entire domain of the primary sensory neuron, including the soma, peripheral and central axons and terminals. Further, available experimental evidence indicates that, in addition to the peripheral receptive endings, propagating action potentials may develop in peripheral (and central) axonal branches of nociceptive primary sensory neurons through the activation of the TRPV1 and ANO1 ion channels. Decreased pH and increased temperature, cardinal signs of tissue inflammation, can readily activate axonal TRPV1 receptors and elicit another dominant trait of the inflammatory process: pain. The generation of ectopic action potential through activation of axonal TRPV1 receptors and ANO1 could contribute to the mechanisms of neuropathic pain evoked by the altered tissue microenvironment around injured peripheral nerves. In light of the experimental observations summarized above, the whole domain of the nociceptive primary sensory neuron is equipped with the nociceptive ion channels TRPV1 and ANO1 and has functional capacities that are closely similar to that of the peripheral nociceptive nerve terminal. Therefore, we propose to term this particular class of capsaicin-sensitive primary sensory neurons as the *chemosensitive nociceptors*, since this term better describes the unique functional capabilities and diverse response characteristics of these neurons.

## Figures and Tables

**Figure 1 ijms-26-10350-f001:**
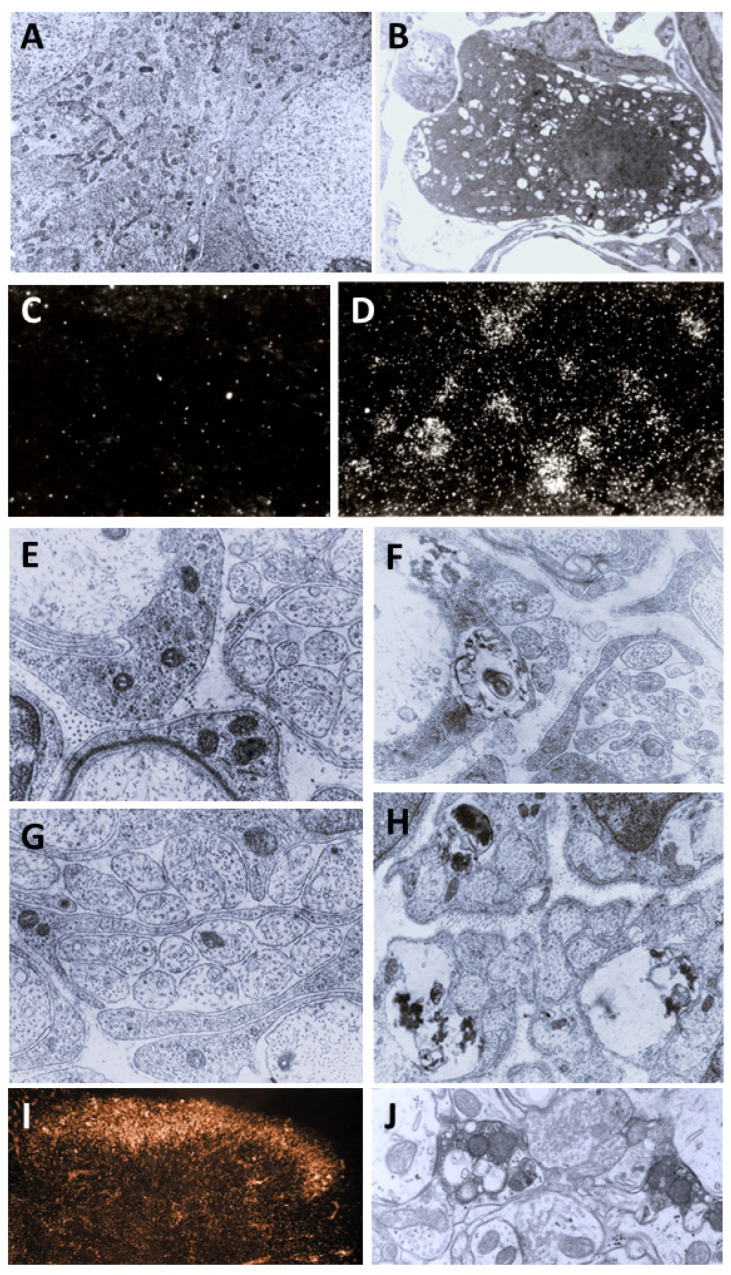
Morphological and cytochemical changes in the domain of primary sensory neurons following neonatal treatment with capsaicin (50 mg/kg, s.c.). (**A**): Electron micrograph of type A and B primary sensory neurons from a spinal ganglion of a 2-day-old rat. (**B**): Electron micrograph of a B-type sensory ganglion neuron, showing clear signs of degeneration: increased nuclear and cytoplasmic osmiophilia, swelling of perikaryal organelles (mitochondria, Golgi cisterns and endoplasmic reticulum) and disappearance of the nuclear membrane 2 h after injection of capsaicin to a 2-day-old rat. (**C**,**D**): Autoradiographic localization of ^45^Ca^2+^ in Gasserian ganglia of control (**C**) and capsaicin-treated (**D**) 2-day-old rats, 25 and 20 min after administration of ^45^Ca^2+^ and capsaicin (**D**) or vehicle (**C**), respectively. Small neurons show marked accumulation of ^45^Ca^2+^ after injection of capsaicin. E-H: Electron micrographs showing details of cervical dorsal roots (**E**,**F**) and saphenous nerves (**G**,**H**) of 2-day-old control rats (**E**,**G**), and rats treated with capsaicin 4 h prior to sacrifice (**F**,**H**). Note unmyelinated axons showing marked osmiophilic degeneration after capsaicin treatment. (**I**): Light microscopic photograph of the spinal cord dorsal horn, showing massive degeneration argyrophilia of axon terminals in the superficial laminae I-II: in particular, the substantia gelatinosa (SG). (**J**): Electron micrograph showing degenerating osmiophilic axon terminals in the substantia gelatinosa of the spinal dorsal horn, 8 h after administration of capsaicin to a 2-day-old rat. (Reproduced from Jancsó G., Morphology and function of chemosensitive primary sensory neurons. Dissertation. Hungarian Academy of Sciences, 1981).

## Data Availability

The photomicrographs of Figure 1 were reproduced from the illustrations of the PhD thesis (1981) of Gábor Jancsó, with his explicit permission.

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
