# Peer review of "Morphological Correlates of TRPV1 Agonist-Induced Activation and Defunctionalization of Nociceptor Neurons"

_ijms, 2025, doi:10.3390/ijms262110350_

Round 1

Reviewer 1 Report

Comments and Suggestions for Authors

This review is well written, however, I recommend adding a few more points to make it better review paper.

The review offers a comprehensive overview of the various morphological changes and functional characteristics associated with TRPV1 receptors, effectively helping readers understand the role of TRPV1. It discusses how TRPV1 agonists may contribute to the treatment of chronic and neuropathic pain, presenting practical implications for clinical use. The clear explanation of the relationship between morphological changes and functional alterations enhances the understanding of the mechanisms underlying TRPV1 receptor action. Additionally, the inclusion of various experimental techniques, such as electron microscopy and immunohistochemistry, bolsters the reliability of the physiological evidence presented regarding TRPV1 functions.

However, I suggest a few points to further enhance the review:

  1. The explanation of the mechanisms of TRPV1 agonists is somewhat general and lacks specific details about the intracellular pathways involved in their action. It would be beneficial to add a dedicated section that focuses on the intracellular signaling pathways activated by TRPV1 agonists, including detailed discussions on calcium signaling, protein kinase activation, and downstream effects on gene expression.
  2. While there is some discussion on the differences in TRPV1 expression between rodents and humans, the analysis of how these inter-species differences may affect the generalizability of the findings is insufficient. A more detailed examination of how TRPV1 expression varies between species and the implications of these differences on the interpretation of experimental results and their applicability to human conditions would strengthen the review.
  3. The review currently has a restricted discussion on specific pathological conditions related to TRPV1, indicating a need for more research on its role in various disease contexts. A deeper exploration of the underlying mechanisms by which TRPV1 contributes to disease processes—such as inflammation, pain signaling, and neuronal plasticity—would be valuable.
  4. There is a lack of discussion regarding the nerve regeneration processes induced by TRPV1 agonists, which limits the understanding of recovery mechanisms in damaged nerves. Including this topic would provide a more holistic view of TRPV1's role in both pain and recovery.
  5. Incorporating diagrams or flowcharts to visually represent the signaling pathways involved or the broad functional roles of TRPV1 could clarify complex interactions and enhance reader understanding of the mechanisms at play.

Author Response

Thank you very much for taking the time and devotion to review this manuscript. Please find the detailed responses below and the corresponding revisions/corrections in track changes in the re-submitted files.

Comments 1: The explanation of the mechanisms of TRPV1 agonists is somewhat general and lacks specific details about the intracellular pathways involved in their action. It would be beneficial to add a dedicated section that focuses on the intracellular signaling pathways activated by TRPV1 agonists, including detailed discussions on calcium signaling, protein kinase activation, and downstream effects on gene expression.

Response 1: Thank you for your comment. Our review focuses on the morphological changes which can be observed in association with TRPV1 activation by vanilloids. However, to comply with your suggestion, we have inserted a new paragraph (Chapter 3., p. 6.) on "Morphological correlates of activation of CPSNs" which touches upon points raised in your comment.

Comments 2: While there is some discussion on the differences in TRPV1 expression between rodents and humans, the analysis of how these inter-species differences may affect the generalizability of the findings is insufficient. A more detailed examination of how TRPV1 expression varies between species and the implications of these differences on the interpretation of experimental results and their applicability to human conditions would strengthen the review.

Response 2:
As mentioned above, the main focus of our manuscript concerns the morphological changes induced by vanilloids. We believe that a detailed discussion of the species-dependent expression of TRPV1 and its role in human pathologies is beyond the scope of this review.

Comments 3: The review currently has a restricted discussion on specific pathological conditions related to TRPV1, indicating a need for more research on its role in various disease contexts. A deeper exploration of the underlying mechanisms by which TRPV1 contributes to disease processes—such as inflammation, pain signaling, and neuronal plasticity—would be valuable.

Response 3: We agree, but feel that discussion of human diseases in the context of TRPV1 function is beyond the framework of this review.

Comments 4: There is a lack of discussion regarding the nerve regeneration processes induced by TRPV1 agonists, which limits the understanding of recovery mechanisms in damaged nerves. Including this topic would provide a more holistic view of TRPV1's role in both pain and recovery.

Response 4: Thank you for drawing our attention to this significant aspect of TRPV1 function. There is ample evidence for the selective degeneration of CPSNs by vanilloid compounds. However, a putative role of TRPV1 in neural regeneration has also been revealed under certain experimental conditions. We have now included these studies. (Chapter 4: page 9. from line 360-374)

Comments 5: Incorporating diagrams or flowcharts to visually represent the signaling pathways involved or the broad functional roles of TRPV1 could clarify complex interactions and enhance reader understanding of the mechanisms at play.

Response 5: We agree, but feel that discussion of signaling pathways relating to TRPV1 functions is beyond the framework of this review. However, since this review is a part of a special iussue (TRP Channels for Pain, Itch and Inflammation Relief: 2nd Edition), these subject areas will  be covered by other contributions.

Reviewer 2 Report

Comments and Suggestions for Authors

Dear authors,
The manuscript aims to review the morphological correlates of TRPV1 agonist–induced activation and defunctionalization in nociceptor neurons. It usefully compiles classic and recent data (ultrastructural changes, IENF loss, peptide depletion, spinal “capsaicin gap”) and introduces the broader construct of “chemosensitive nociceptors.” However, the narrative often extends beyond the stated focus, devotes disproportionate space to tangential topics, and has some other issues that need to be addressed. 

Thank you for choosing IJMS for your submission. We appreciate the effort and the valuable synthesis you are building. However, the issues outlined above require extensive revision before the manuscript can be considered for publication. We encourage you to address these points thoroughly and look forward to reviewing a strengthened, revised version.

Author Response

Response to Reviewer 2 and Comments

Thank you very much for taking the time to review this manuscript. Please find the detailed responses below and the corresponding revisions/corrections in track changes in the re-submitted files!

General recommendation:

Remove (or substantially reduce) sections on migraine, obesity, diabetes, adriamycin, purinergic signaling, and the “trigeminal nocisensor complex” unless directly tied to TRPV1 agonist exposure (capsaicin/RTX) and morphological readouts.

Response: In accord with the Reviewer suggestion, we have removed these sections.

Comments 1: Broad CTB/GM1/GSL metabolism narrative not tied to agonist-induced morphology; retain only the CTB reinterpretation that prevents misattributing A-fiber sprouting and any direct agonist-linked GSL/morphology evidence.

Response 1: We have substantially rewritten the sections dealing with this subject. We have focused on observations which clarified the true nature of changes in the spinal distribution of CTB-labelled primary afferents. We have kept the information which pointed to the possible functional role of ganglioside GM1 in the regulation of capsaicin sensitivity and TRPV1 expression. (see revised Chapter 5 from p. 12)

Comments 2:  Speculative tumor microenvironment paragraphs lacking concrete agonist-linked peripheral nerve morphology.

Response 2: We have added new data supporting the role of sensory nerves and, in particular TRPV1 receptor in tumor growth and progression. (p. 16, lines 622-28)

Comments 3: Imbalance: strong for defunctionalization, weak for activation (morphology) Add acute structural correlates of activation (beyond CGRP release, ERK/c-Fos, localization). If evidence is scarce, state this explicitly as a gap.

Response 3: We have added a new section to the manuscript describing the histomorphological changes associated with activation of CPSNs. (Chapter 3. p. 6. from line 214)

Comments 4: Temper statements implying uniform TRPV1/ANO1 distribution/function across compartments; add caveats on expression density, trafficking, and context. Reframe tumor–perineural microenvironment claims as speculative or remove without direct morphological evidence.

Respone 4: We have added new information on the possible role of the ANO1 ion channel in the TRPV1-mediated nociceptive functions at the cellular and the behavioral levels. (p. 16, lines 608-621)

Comments 5: Define and consistently use the primary construct (e.g., chemosensitive nociceptors) and map it explicitly to TRPV1+/TRPA1+/non-peptidergic subsets. Align usage of CPSN, “capsaicin- sensitive,” and “TRPV1-expressing.”

Response 5: We have dealt with this issue and now use the term CPSN throughout the manuscript. As discussed in the last section of the manuscript, we suggest the use of the term "chemosensitive nociceptor" which, in our opinion, describes more properly the functional traits and properties of these particular class of nociceptive neurons. (also see: chapter 2.; p.2, from line 49).

Comments 6: Add a comparative morphological analysis (dose/exposure, compartment, time course, structural readouts, outcomes) and discuss mechanisms (e.g., Ca2+ load, kinetics, axonal access) that reconcile RTX analgesia without degeneration vs. capsaicin-linked degeneration.

Response 6: We have added detailed information and comparative data on the dosage, time course and structural changes induced by capsaicin and resiniferatoxin, respectively. We have also discussed and raised some points which may explain the differing findings on the structural consequences of capsaicin and resiniferatoxin treatments.  (Chapter 4.  p. 8, lines 300-316)

Comments 7: Either integrate CTB/GM1/ganglioside content tightly with agonist-induced morphological outcomes (show how ganglioside changes are agonist-linked) or remove.

Response 7: We have rephrased the parts of the manuscript dealing with this issue. (Chapter 5, from p. 12)

Comments 8. Provide a short “What’s new in this review” paragraph, in the introduction. Explain what is the gap in knowledge your review is trying to cover and how it can be useful for the reader. Explain how it differs from other similar reviews. State the type of review in the Introduction – e.g. “This narrative review....”

Response 8:  We have rewritten the introductory part of the manuscript to address these issues. (see Chapter 1. p. 1) 

Comments 9: (Insert) Table 1: Morphological correlates of defunctionalization (IENF loss, peptide depletion, EM degeneration, capsaisin gap; model, dose, timeline).

Respone 9: We have added Table I. summarizing the morphological findings following the administration of capsaicin through various routes (pp. 10-11)

Comments 10: (Insert) Table 2: Putative morphological correlates of activation (if sparse, state as gap).

Response 10: We have added a new chapter "Morphological correlates of activation of CPSNs" to deal with this issue. (from p. 6., line 214)

Comments 11: (Insert) Table 3: Capsaicin vs. RTX (methods, compartments, outcomes).

Response 11: We have added additional information on capsaicin vs resiniferatoxin. (Chapter 4.  p. 8, lines 300-316)

Comments 12: Temper over-strong generalizations about TRPV1/ANO1 distribution and capacity across all compartments; add caveats on expression density, trafficking, and state-dependence.

Response 12:  The section on TRPV1/ANO1 has been rephrased by adding new information. (p. 16, lines 609-621)

Comments 13: The English requires revision to improve clarity, grammar, and style.

Response 13:  We have revised large parts of the text.

Comments 14: Use journal template consistently. Number the chapters.

Response 14: We have made the necessary corrections and numbered the chapters.

Comments 15: Ensure consistent tense and British vs. American English.

Response 15: We have made the necessary corrections.

Comments 16: Rewrite ambiguous sentences (e.g., the “selective analgesia...” line), fix duplicated lines/typos, and standardize abbreviations at first use (IENF, FRAP/TMP, etc.). Normalize hyphenation (“perineural,” “chemodenervation,” “immunoreactive”).

Response 16: Thank you for pointing to these issues. We have fixed them.

Comments 17: Cross-check all percentages and time frames for internal consistency (e.g., “~30–35% unmyelinated fiber loss” vs. “~50% depletion”—align wording and cite methods).

Response 17: We have checked these percentage data and rewrote this section.

Reviewer 3 Report

Comments and Suggestions for Authors

This review is clearly the result of a lot of work. It’s thorough, it pulls together both classic and modern literature, and it gives readers a good overview of how TRPV1 agonists affect nociceptors. The historical background is also interesting and shows the authors’ deep familiarity with the field.

At the same time, the paper is very long and sometimes hard to follow. Some sentences go on forever, there are quite a few repetitions, and it’s not always clear what is well established and what is still a hypothesis. A few technical mistakes and typos also distract from the overall quality. With some trimming, clarification, and updates, the manuscript could become a strong and very useful reference.

Major points

  • Page 1, lines 22–26: The introduction of “chemosensitive nociceptor neuron” is one huge sentence. It would read much better split into two or three.
  • Page 2, lines 46–57: The historical section is a bit too narrative. Shorten it and make clear which of these old findings still hold up today.
  • The term chemosensitive nociceptor neuron (Page 1, lines 25–26) should be backed with a proper citation. Right now, it feels like the concept just appears out of nowhere.
  • Page 2, lines 59–71: The description of mitochondrial changes and substance P depletion mixes hard evidence with speculation. Please separate what’s solid data from what’s hypothetical.
  • Page 5, lines 160–176: You compare TRPV1-positive neurons in mouse, rat, and human DRGs. That’s important, but the methods used (IHC, RNA-seq, etc.) are not mentioned. This could mislead readers.
  • Page 9, lines 336–350: Phrases like “several factors may enhance severity” are too general. Try to include some numbers or concrete clinical findings.
  • Page 9, lines 369–375: The “nocisensor complex” is an interesting idea, but the reference given is too general. Please cite the first study that described this concept.
  • Page 10, lines 410–417: The text mentions reduced TRPV1-positive fibers in diabetic rats, but doesn’t explain how this was measured. Readers need at least a short note about the method (IHC, quantification).
  • Pages 11–12, lines 487–507: This section is heavy on abbreviations. Some (like GCS) appear without explanation. Please define them the first time you use them.

Minor issues

  • Page 7, line 243: The text cites “unpublished data.” That’s not suitable for a review and should be removed.
  • The references lean heavily on older papers (1960s–70s). Please add more recent reviews (2020–2024) to balance this.
  • Long citation blocks (e.g. [19,33–44]) are hard to digest. Better to highlight only the most relevant ones.

Recommendation

I would suggest major revisions. The paper has strong content, but it needs clearer writing, some trimming, more recent references, and a few technical fixes. Once that’s done, it could be a valuable and widely read review in the TRPV1 field.

Author Response

Thank you very much for taking the time to review this manuscript. Please find the detailed responses below and the corresponding revisions/corrections in track changes in the re-submitted files.

Comments 1: Page 1, lines 22–26: The introduction of “chemosensitive nociceptor neuron” is one huge sentence. It would read much better split into two or three.

Response 1: Thank you for your comment. We have rephrased these sentences (p. 1, lines 32-36)

Comments 2: Page 2, lines 46–57: The historical section is a bit too narrative. Shorten it and make clear which of these old findings still hold up today.

Response 2: Thank you for your comment. We have omitted some sentences. (p. 2. lines 57-65)

Comments 3: The term chemosensitive nociceptor neuron (Page 1, lines 25–26) should be backed with a proper citation. Right now, it feels like the concept just appears out of nowhere.

Response 3: We have dealt with this issue and now use the term CPSN throughout the manuscript. As discussed in the last section of the manuscript, we suggest the use of the term "chemosensitive nociceptor" which, in our opinion, describes more properly the functional traits and properties of these particular class of nociceptive neurons. (also see Chapter 2, p. 2. lines 53-57)

Comments 4: Page 2, lines 59–71: The description of mitochondrial changes and substance P depletion mixes hard evidence with speculation. Please separate what’s solid data from what’s hypothetical.

Response 4:  This section describes early morphological findings following administration of capsaicin in the adult rat. These findings have been repeatedly confirmed in later years. As to the speculation on FRAP and substance P, I would like to refer to the original publication of Jessell and colleagues who described how they came to the idea to investigate possible changes in the localization of substance P in the spinal cord after capsaicin. They wrote: "The striking similarity between the distribution of substance P and the fluoride-resistant acid phosphatase in spinal cord led us to investigate the possibility that capsaicin-induced desensitization may be mediated by an action on substance P-containing primary afferent terminals in the substantia gelatinosa."

Comments 5: Page 5, lines 160–176: You compare TRPV1-positive neurons in mouse, rat, and human DRGs. That’s important, but the methods used (IHC, RNA-seq, etc.) are not mentioned. This could mislead readers.

Response 5: Thank you for this comment. We have added the necessary information. (p5. 160-167)

Comments 6: Page 9, lines 336–350: Phrases like “several factors may enhance severity” are too general. Try to include some numbers or concrete clinical findings.

Response 6: This section of the manuscript has been removed.

Comments 7: Page 9, lines 369–375: The “nocisensor complex” is an interesting idea, but the reference given is too general. Please cite the first study that described this concept.

Response 7: This section of the manuscript has been removed.

Comments 8: Page 10, lines 410–417: The text mentions reduced TRPV1-positive fibers in diabetic rats, but doesn’t explain how this was measured. Readers need at least a short note about the method (IHC, quantification).

Response 8: This section of the manuscript has been removed.

Comments 9: Pages 11–12, lines 487–507: This section is heavy on abbreviations. Some (like GCS) appear without explanation. Please define them the first time you use them.

Response 9: We have made the necessary corrections. (see revised Chapter 5., p. 13)

Comments 10: Page 7, line 243: The text cites “unpublished data.” That’s not suitable for a review and should be removed.

Response 10: This has been removed.

Comments 11: The references lean heavily on older papers (1960s–70s). Please add more recent reviews (2020–2024) to balance this.

Response 11: We have added some new papers.

Comments 12: Long citation blocks (e.g. [19,33–44]) are hard to digest. Better to highlight only the most relevant ones.

Response 12: We have made the necessary corrections.

Round 2

Reviewer 2 Report

Comments and Suggestions for Authors

Dear authors,

Thank you for your thorough and thoughtful revisions. The manuscript has been significantly improved in clarity. I find that all major and minor concerns raised in the first review have been satisfactorily addressed. The revised version presents a clear, well-structured, and scientifically sound contribution to the understanding of TRPV1 agonist-induced nociceptor activation and defunctionalization.

I have no further comments — I consider the manuscript suitable for publication in IJMS in its current form.